# Mitochondrial Homeostasis and Cellular Senescence

**DOI:** 10.3390/cells8070686

**Published:** 2019-07-06

**Authors:** Panagiotis V.S. Vasileiou, Konstantinos Evangelou, Konstantinos Vlasis, Georgios Fildisis, Mihalis I. Panayiotidis, Efstathios Chronopoulos, Panagiotis-Georgios Passias, Mirsini Kouloukoussa, Vassilis G. Gorgoulis, Sophia Havaki

**Affiliations:** 1Molecular Carcinogenesis Group, Department of Histology and Embryology, Medical School, National and Kapodistrian University of Athens, 75 Mikras Asias Str., 11527 Athens, Greece; 2Department of Anatomy, Medical School, National and Kapodistrian University of Athens, 75 Mikras Asias Str., 11527 Athens, Greece; 3Nursing School, National and Kapodistrian University of Athens, 123 Papadiamantopoulou Str., 11527 Athens, Greece; 4Department of Applied Sciences, Northumbria University, Newcastle upon Tyne, NE1 8ST Newcastle, UK; 5Second Department of Orthopaedics, Medical School, National and Kapodistrian University of Athens, 75 Mikras Asias Str., 11527 Athens, Greece; 6Faculty Institute for Cancer Sciences, Manchester Academic Health Sciences Centre, University of Manchester, Manchester MP13 9PL, UK; 7Biomedical Research Foundation of the Academy of Athens, 4 Soranou Ephessiou Str., 11527 Athens, Greece; 8Center for New Biotechnologies and Precision Medicine, Medical School, National and Kapodistrian University of Athens, 75 Mikras Asias Str., 11527 Athens, Greece

**Keywords:** cellular senescence, mitochondria, mitostasis, mitochondrial dynamics

## Abstract

Cellular senescence refers to a stress response aiming to preserve cellular and, therefore, organismal homeostasis. Importantly, deregulation of mitochondrial homeostatic mechanisms, manifested as impaired mitochondrial biogenesis, metabolism and dynamics, has emerged as a hallmark of cellular senescence. On the other hand, impaired mitostasis has been suggested to induce cellular senescence. This review aims to provide an overview of homeostatic mechanisms operating within mitochondria and a comprehensive insight into the interplay between cellular senescence and mitochondrial dysfunction.

## 1. Introduction

Cellular senescence is part of a range of cell responses towards extrinsic and/or intrinsic noxious insults that challenge homeostasis, mainly genome and proteome integrity (Figure 1) [1]. The senescent cell is a stressed or damaged, yet viable, cell that has entered a non-proliferative state while still remaining metabolically active. Historically, the first condition described leading to senescence was exhaustion of replication potential due to serial passaging leading to telomere attrition [2]. Below a critical length of telomere, a deoxyribonucleic acid (DNA) damage response is triggered imposing a type of senescence termed replicative senescence (RS). Except for telomere attrition, a wide range of other telomere-independent stimuli, such as oxidative stress, activated oncogenes (termed oncogene induced senescence/OIS), irradiation, genotoxic drugs, cell–cell fusion, epigenetic modifiers, or perturbed proteostasis, have been recognized as powerful inducers of cell senescence. Senescence can also be induced by failure to repair DNA damage [1,3,4]. Senescence imposed by telomere-independent stimuli is more acute and is known as stress induced premature senescence (SIPS) [4,5,6,7,8]. Mechanistically, several molecular pathways have been implicated that often depend on the nature of the initiating event and/or cell type [9,10]. Two best studied molecular axes involve p53/p21^WAF1^ and Rb-p16^INK4A^ that can also reinforce senescence via a ROS-dependent positive feedback mechanism [5,11,12]. Notably, the p53/p21^WAF1^ pathway has been suggested to initiate the senescence response, followed by the action of p16^INK4A^ to maintain this condition [13].

A variety of cellular and molecular hallmarks of senescence have been so far identified, including resistance to apoptosis, morphological and structural features, epigenetic alterations, chromatin rearrangement, and a modified transcriptome program [9,14]. Indeed, senescent cells are known for their increased secreting activity [5]. Particularly, they carry out a complex pro-inflammatory response known as senescence-associated secretory phenotype (SASP), which is mediated by the transcription nuclear factor-κB (NF-κB) and includes the secretion of a spectrum of pro-inflammatory factors, such as interleukins, chemokines, growth factors, proteases, cell surface molecules, and extracellular matrix degrading proteins, that influence the surrounding microenvironment. Respectively, the constituents of SASP act in an autocrine and paracrine manner contributing in various developmental programs or pathophysiological conditions [4,5,6,9,15,16]. Closely related with SASP, senescent cells also exhibit apparent alterations of cellular metabolism, corresponding to abnormalities in morphology, mass, and functionality of their organelles [17].

At this point, and by virtue of their central bioenergetic role and their involvement in other physiological processes such as redox signaling, mitochondria enter the scene as potential key players during cellular senescence [18,19]. Cumulative data support this notion. Mitochondrial oxidative phosphorylation (OXPHOS) deterioration has been reported to be primarily involved in the early stages of cellular senescence, using diverse cellular senescence models [20,21,22,23,24,25]. Senescent cells are characterized by increased production of reactive oxygen species (ROS), mainly attributed to dysfunctional mitochondria [26]. Indeed, in already senescent cells, mitochondrial ROS can aggravate cellular senescence by enhancing the DNA damage and the DNA damage response signaling pathway (DDR) [11]. Noteworthy, mitochondrial deoxyribonucleic acid (mtDNA) is highly vulnerable to ROS due to proximity to the generation site, whilst damaged mtDNA in turn, impairs OXPHOS function, thus further enhancing ROS release [17]. Furthermore, senescent cells exert massive metabolic changes related to mitochondrial metabolites [e.g., oxidized to reduced form of nicotinamide adenine dinucleotide ratios (NAD^+^/NADH) or tricarboxylic acid (TCA) cycle metabolites], and dynamics (namely fusion, fission and mitophagy) [18,19]. Additionally, mitochondrial biogenesis is up-regulated during senescence [11,27]. Notably, despite the increased mitochondrial pool, the overall adenosine triphosphate (ATP) production by oxidative phosphorylation is reduced during senescence [28]. Furthermore, mitochondria of senescent cells show decreased membrane potential, accelerated ROS production and are prone to leakage of mitochondrial enzymes [29,30].

Not only is mitochondrial dysfunction an epiphenomenon of senescence, but also dysfunctional mitochondria can indeed drive the senescent phenotype. Perturbation of mitochondrial homeostasis promotes the establishment and maintenance of cellular senescence through various mechanisms including excessive mitochondrial ROS production, imbalanced mitochondrial dynamics, electron transport chain defect, bioenergetics imbalance and increased 5’ adenosine monophosphate-activated protein kinase (AMPK) activity, altered mitochondrial metabolite profile (e.g., NAD^+^), and dysregulated mitochondrial calcium homeostasis [31]. These mitochondrial signals trigger p53/p21^WAF1^ and/or Rb-p16^INK4A^ pathways, ultimately leading to cellular senescence and stabilizing cell-cycle arrest [11,31,32,33,34]. A number of studies indicate that mitochondrial-derived ROS can accelerate telomere shortening, thus causing premature senescence [29], triggering paracrine senescence [35], or inducing and maintaining senescence through sustained DNA damage response [11,29,36]. Strikingly, clearance of mitochondria negatively impacts the development of many senescence-associated features, including the SASP, while maintaining cell-cycle arrest [37]. Recently, the induction of mitochondrial dysfunction was reported to generate a distinct (i.e., mainly in terms of SASP) type of senescence termed mitochondrial dysfunction-associated senescence (MiDAS) [38].

Apparently, a growing body of evidence underscores a bidirectional link between cellular senescence and these multifaceted organelles. This interplay seems to be best described as a vicious circle, involving a number of feedback loops between the players, rather than a linear cause and effect relationship [19]. Notably, the implication of mitochondria in the context of cellular senescence extends far beyond their contribution in ROS production and oxidative stress. In view of recent outstanding findings regarding the role of mitochondria in cellular senescence, herein we sought to present an overview of mitochondrial homeostatic mechanisms along with evidence implicating mitostasis aberrations in cellular senescence or vice versa.

## 2. Mitostasis: An Overview of the Mitochondrial Genome and Proteome Maintenance Mechanisms

Mitostasis is a term used to encompass all the mechanisms implicated in the maintenance of normal mitochondrial function. It refers both to genome and proteome integrity of mitochondrion.

### 2.1. Mitochondrial Genome Maintenance Mechanisms

Mammalian mitochondria biogenesis and function require the coordinated action of two genomes: nuclear and mitochondrial [39].

Mammalian mtDNA is a small, adenine/thymine-rich, circular molecule consisting of 16,569-base pairs [40]. Its small size confers two benefits: it enhances rapidity and facilitates accuracy of replication [41]. MtDNA contains 37 genes coding for 2 ribosomal nucleic acids, 22 transfer RNAs, and 13 essential protein subunits of the oxidative phosphorylation system. Each organelle contains two to five copies of mtDNA, therefore each cell has thousands (approximately 1000–l0,000) apparently identical copies of mtDNA [42]. Despite its small size and due to its polyploid nature, mtDNA can represent approximately 1% of the total DNA in some cells [43].

The replication of mtDNA is not limited to the S phase, but occurs throughout the cell cycle. Of interest, two modes of mtDNA replication operate in mammalian; the initially described, “orthodox”, strand-asymmetric mechanism [44], and the unidirectional, synchronous leading- and lagging-strand replication cells [45].

A number of surprising features characterizing the mitochondrial genome have come to light, such as dense gene packing, low methylation levels, relaxed codon usage, and a variant genetic code [40,46,47]. In mammalian mtDNA, the addition of a third DNA strand (∼0.5 kb), termed “7S DNA”, forms the displacement-loop (d-loop), a short triple-stranded, non-coding, regulatory region of mtDNA responsible for transcription and replication initiation by the mitochondria-specific polymerase-γ (pol γ) [48,49]. In addition, d-loop has been implicated in protein recruitment, mtDNA organization and metabolism, as well as dNTP pools maintenance throughout the cell cycle [50,51,52]. Importantly, many but not all molecules of mtDNA bear this third strand of DNA. In fact, the abundance of 7S-DNA varies greatly between species and cell type, being present on 1–65% of mtDNA molecules [53,54]. Strikingly, other molecules contain RNA as the third strand. The RNA of these R-loops is similar in length and location to the d-loop and is complementary to 7S DNA. Of clinical relevance, in cells with a pathological variant of ribonuclease H1 (an enzyme that degrades RNA hybridized to DNA) associated with mitochondrial disease, R-loop numbers are low and there is mitochondrial DNA aggregation, strongly suggesting a role for the R-loop in mtDNA organization and segregation [55].

MtDNA is packaged into protein–DNA complexes called nucleoids [56,57]. The main DNA packaging protein of nucleoids is the mitochondrial transcription factor A (TFAM), a member of the high-mobility group (HMG) of proteins [58,59]. Other factors exerting central role in the maintenance of the mitochondrial genome’s integrity are the nuclear respiratory factors 1 and 2 (NRF 1/2), which are implicated in the transcriptional control of mtDNA, the peroxisome proliferator-activated receptor gamma co-activator one alpha (PGC1α), which stimulates mitochondrial biogenesis in the basis of cellular energy metabolism regulation, as well as sirtuins (SIRT) [60,61,62]. Mitochondrial sirtuins—SIRT3, SIRT4, and SIRT5—are NAD^+^-dependent deacetylases, deacylases, and ADP-ribosyl transferases. Their enzymatic activity is indirectly (through NAD^+^) linked to the metabolic state of the cell. Importantly, they also regulate non-metabolic aspects of mitochondrial biology, thus ensuring that mitochondrial homeostasis is achieved during stress conditions [63].

The main polymerase functioning within mitochondria is polymerase γ (Pol γ), a heterotrimer comprised of one pol γ catalytic subunit (p140), which exerts a DNA polymerase activity, a 3′-5′ exonuclease activity and a 5′-deoxyribose lyase activity, and two accessory subunits (p55). Contrary to the high nucleotide selectivity and exonucleolytic proofreading of the isolated pol γ catalytic subunit, p55 dimeric exerts reduced fidelity of DNA replication by promoting extension of mismatched DNA termini [64]. Importantly, the general notion that pol γ is uniquely responsible for replication and repair of mitochondrial DNA, has been recently challenged, since several polymerases are now proposed to be present within these organelles [65]. For example, it has been demonstrated that Polβ is involved in mtDNA maintenance. At least in some tissues, Polβ interacts with nucleoid proteins such as TWINKLE helicase, mitochondrial single-strand DNA-binding protein 1 (SSBP1 or mtSSB), and TFAM, thus contributing to mtDNA repair machinery [66]. Another example of such a player is PrimPol, a polymerase which also acts as a primase, having roles in both nuclear and mitochondrial DNA maintenance. PrimPol identified in human mitochondria exerts de novo DNA synthesis capability and oxidative lesions tolerance. Moreover, it seems to play additional roles in the repair of damaged DNA in the absence of ongoing replication [67,68]. Nevertheless, the exact role of all polymerases identified within mitochondria is not yet clear [65].

The integrity of mtDNA, which is crucial for mitostasis, is maintained by multiple DNA repair pathways and through the selective degradation of irreparable or heavily damaged DNA. Indeed, stability of the mitochondrial genome is fulfilled through a 3-level defense system, including (a) the architectural organization of mtDNA, (b) DNA repair mechanisms that are activated within mitochondria when mtDNA damage occurs, and (c) the cleavage of damaged mtDNA through mitochondrial dynamic processes [69]. Importantly, our knowledge regarding DNA repair pathways operating within these multifaceted organelles has been expanding during the last decades, from the inceptive belief of no available repair mechanisms, through the subsequent identification of a limited repair repertoire, to the recent and constantly evolving awareness of a sufficient and vigorous “arsenal” against mitochondrial genome damage [70]. Except for the direct reversal (DR) of certain lesions and short-patch base excision repair (BER) [71,72,73], mitochondria also exert long-patch BER activity and translesion synthesis (TLS) capacity for the repair of single-strand breaks, as well as homology recombination (HR), non-homologous (NHEJ) and microhomology-mediated end-joining (MMEJ) activities for the repair of double-strand lesions [67,74,75,76,77,78,79,80]. Additionally, a novel mismatch repair (MMR) pathway, distinctive from the nuclear one, has been shown to be also present within mitochondria [81,82]. However, the level of proficiency of each one of these repair mechanisms, regarding their intra-mitochondrial functionality, has not been fully elucidated and remains to be further studied in order to characterize key players and regulators involved, both in vitro and in vivo. Collectively, with the exception of nucleotide excision repair (NER) and Fanconi anemia (FA) pathways which have not yet been identified within mitochondria, it appears that a broad range of DNA repair mechanisms that operate in the nucleus contribute also to the integrity of the mitochondrial genome. To date, the only hint regarding the NER pathway in the mitochondria is the localization of the transcription-coupled NER proteins CSA and CSB (Cockayne syndrome proteins) to mitochondria upon oxidative stress [83]. Interestingly, recent evidence supports that multiple proteins in the FA pathway are involved in the suppression of inflammasome activation by decreasing mitochondrial ROS production, and are required for mitophagy (clearance of damaged mitochondria) through interaction of FANCC (Fanconi anemia complementation group C) protein with Parkin, thus contributing to mitochondrial and cell homeostasis [84].

### 2.2. Mitochondrial Proteome Maintenance Mechanisms

A wide range of proteins are involved in the organization, regulation and replication of the mitochondrial genome and the assembly of these multifaceted organelles.

Proteomic studies, driven by large-scale approaches, including in-depth protein mass spectrometry, microscopical, computational and integrative machine learning methods, revealed that mitochondria contain approximately 1000 (in yeast) to 1500 (in humans) different proteins [85,86,87]. From a functional perspective, mitochondrial and mitochondrial-associated proteins are mainly distributed/classified in those involved in energy metabolism (≈15%), protein synthesis, transport, folding and turnover functions (≈23%), and genome maintenance and transcription (12%) [88]. Other mitochondrial functions, including structural, signaling and redox processes, transport of metabolites, as well as iron, amino-acid and lipid metabolism, occupy the remaining 30% of the mitochondrial protein armament. Of note, for more than 19% of mitochondrial proteins, no reliable information on their function is available [85,89].

Most mitochondrial proteins are synthesized on cytosolic ribosomes and must be imported across one or both mitochondrial membranes [90]. Only 13 (about 1%) from the total number of peptides that compose the mitochondrial proteome are encoded by the mitochondrial DNA and synthesized in the mitochondrial matrix, while the remaining 99% of the mitochondrial proteins are encoded by nuclear genes [85]. Thus, the larger part of the mitochondrial proteins needs to travel in an unfolded state from the cytosol into the mitochondrion [86,91,92]. Trafficking and import of mitochondrial precursor proteins (pre-proteins) is mainly mediated by two mitochondrial translocases, namely the Translocase of the Outer Membrane (TOM) and the Translocase of the Inner Membrane (TIM) complexes [93,94]. Importantly, it has become clear that aberrant routes bypassing the preprotein translocases pathways also exist. In this regard, four principal pathways that direct proteins to their intramitochondrial destination have been so far recognized: the presequence pathway to the matrix and inner membrane, the carrier protein pathway to the inner membrane, the redox-regulated import pathway into the intermembrane space, and the β-barrel pathway into the outer membrane [90].

Proper assembly and quality control of mitochondrial proteins is further monitored and executed by a group of molecular chaperones (also known as “heat shock proteins”) which function in collaboration with a group of proteolytic enzymes (proteases) [94,95,96]. In fact, mitochondria possess their own group of chaperones and proteases stationed in the four compartments of the organelle (i.e., the outer membrane, the intermembrane space, the inner membrane and the matrix) [97,98,99]. These compartment-specific chaperones perform multiple functions important for mitochondria biogenesis and maintenance [100,101]. First, they are essential constituents of the mitochondrial protein import machinery, thus enabling transmembrane trafficking of these macromolecules [102]. Second, molecular chaperones are responsible for proper folding of nascent polypeptides and have a role in intra-mitochondrial protein synthesis [95,103,104]. Third, they protect mitochondrial proteins against denaturation and are actively involved in disaggregation and refolding/remodeling of protein aggregates formed under stress conditions [95]. Of note, an additional specific task for mitochondrial chaperones is their involvement in the maintenance and replication of mitochondrial DNA [105]. The two most dynamic networks of mitochondria chaperones are the mt-Hsp70 (an Hsp70 family member) and the multimeric Hsp60-Hsp10 machineries [90]. The former assists translocation of preproteins across both the outer and inner mitochondrial membranes via an ATP-dependent process, whereas the latter is required for the folding of new protein precursors [106,107]. Chaperone Hsp78 (a member of the ClpB/Hsp104 family) is also implicated in mitostasis, fulfilling an essential role for the respiratory chain reaction and the mitochondrial genome’s integrity under severe stress [108]. In particular, Hsp78 in cooperation with co-chaperones (e.g., Hsp70) drives restoration of the original mitochondrial network/morphology or the translation and synthesis of mitochondrial DNA, upon heat shock [104,109]. Another molecular chaperone identified to be localized in the mitochondrial matrix is TRAP1 (tumor necrosis factor receptor-associated protein 1), a Hsp90-like chaperone, which is a critical regulator of a variety of physiological functions, including cell proliferation, differentiation, and survival [110,111]. Among other tasks, TRAP1 regulates the metabolic shift between oxidative phosphorylation to aerobic glycolysis (a hallmark of cancerous cells’ metabolism, called “Warburg Effect”) [112]. Interestingly, TRAP1 expression is up-regulated in mitochondria of various tumor cells, but is down-regulated in mitochondria of corresponding normal tissues [113]. Furthermore, TRAP1 prevents cell death induced by ROS accumulation or mitochondrial permeability transition pore opening [114,115,116].

The mitochondrial protein quality control surveillance mechanism is further supported by a complex network of mitochondrial proteases, which monitor all four mitochondrial compartments against deleterious accumulation of misfolded, misassembled or unfolded proteins [97]. Among a plethora of enzymes, this group of localized proteases includes: a) the ATP-dependent proteases, namely, the LON protease, the Clp Protease Proteolytic subunit (CLPP) and the presequence protease (PITRM1), located in the matrix, b) the mitochondrial AAA (ATPases Associated with diverse cellular Activities) and PARL (Presenilins-associated rhomboid-like protein) proteases of the inner mitochondrial membrane; and c) the two ATP independent proteases, the ATP23 and HTRA2, and the mitochondrial oligopeptidase M (MEP) which reside in the intermembrane space [94,97,117,118]. Collectively, human mitodegradome consists of at least 25 exclusively mitochondrial components that can be grouped into three different catalytic classes: (a) 2 Cys proteases, (b) 15 metalloproteases and (c) 8 Ser proteases [117]. Depending on their function, location as well as structural and proteolytic characteristics, mitochondrial proteases (mitoproteases) can be divided into two groups. The first group is formed by 20 “intrinsic mitoproteases”, the functional activity of which is mostly performed in the mitochondrion; the second group includes five catalytically deficient but functionally proficient mitochondrial proteins, termed “pseudo-mitoproteases”. Even though these pseudo-mitoproteases lack some key residues for catalysis, they exert a regulatory effect on homologous proteases. A discrete group comprising at least 20 proteases are transiently translocated to mitochondria to perform additional proteolytic activities (mainly related to apoptosis or autophagy), under certain circumstances (i.e., in response to excessive stress) [117]. Importantly, the role of mitoproteases in mitochondrial homeostasis extends far beyond their basic function as proteolytic and degradative enzymes. By ensuring proper protein import, maturation and processing, influencing the half-lives of key regulatory proteins, and activating/deactivating proteins essential for core mitochondrial activities in a highly specific and regulated manner, mitoproteases have been recognized as key regulators of mitochondrial gene expression, mitochondrial biogenesis and dynamics, mitophagy and apoptosis. Furthermore, new evidence highlights the impact of impaired or dysregulated function of mitochondrial proteases in the control of ageing and longevity [119,120,121,122,123,124].

Recently, an additional role for the cytosol-localized ubiquitin-proteasome system (UPS), a key component of the cellular proteostasis network (PN), has begun to emerge regarding mitostasis. Particularly, UPS has been implicated in protein quality control of the mitochondrial outer membrane or protein import into the organelle [125,126,127]. Despite the fact that no specific mitoproteases have been identified so far at the outer mitochondrial membrane, a number of ubiquitin ligases have been found to reside to the cytosolic side of this compartment, including the mitochondrial ubiquitin ligase MITOL [also known as membrane-associated ring finger 5 (MARCH-V)], the mitochondrial E3 ubiquitin protein ligase 1 (MULAN), and the mitochondrial distribution and morphology protein 30 (Mdm30) [128]. Of note, UPS is also involved in mitochondrial fusion and fission [94,129,130,131,132,133,134]. Since the mitochondrial outer membrane accommodates several proteins involved in mitochondrial morphology and dynamics, and given the crucial role of mitochondrial morphology and dynamics for cell cycle progression and/or cell fate, it becomes prevalent how important the protein quality control of this specific mitochondrial compartment is [135,136,137]. Consistent with its contribution in controlling the outer membrane protein quality is the role of UPS in the regulation of the proteome of other mitochondrial compartments, such as the matrix (oligomycin sensitivity-conferring protein/OSCP, component of the mitochondrial membrane ATP synthase), the intramembrane space (endonuclease G), and the inner membrane (Uncoupling Protein-2/UCP2 and Uncoupling Protein-3/UCP3) [138,139,140].

Of great importance, during impaired mitochondrial function and/or instability of the mitochondrial proteome, cells can employ a specific ubiquitin-proteasome mitochondrial stress response known as mitochondrial UPR (UPR^mt^). This mitochondrial stress response mechanism is characterized by the induction of mitochondrial proteostasis machinery (such as mitochondrial molecular chaperones and proteases) as well as anti-oxidant genes to limit damage due to increased generation of reactive oxygen species [141,142]. UPR^mt^ provides a link between mitochondrial survival pathways and the multitasking UPS [94]. In case of irreversible impairment of mitostasis, UPR^mt^ induces outer mitochondrial membrane-associated degradation and/or mitophagy or even apoptosis [94,97].

### 2.3. Mitochondrial Dynamics

Another aspect regarding the maintenance of mitochondrial homeostasis is mitochondrial dynamics, a term used to encompass three main events: fusion, fission, and mitophagy (i.e., selective mitochondrial autophagy) [143,144]. Fusion dilutes and rearranges the matrix content of a damaged mitochondrion (e.g., a mitochondrion containing unfolded proteome or mutated DNA) with a healthy one, whereas fission partitions damaged material to daughter organelles, thus functioning as mitochondrial quality control mechanisms. During cell cycle progression, mitochondria typically elongate in the G1/S phase, in order to ensure greater ATP supply required to sustain cell duplication, and fragment in the G2/M phase to be equally divided to daughter cells as well as to partition damaged material to daughter organelles [145,146,147,148]. A tightly controlled balance between fission and fusion events is required to ensure normal mitochondrial and cellular functions. Notably, the relative rates of fusion and fission mainly define mitochondrial architecture. Furthermore, both these processes are closely related to the biochemical and metabolic cell status [145,149,150].

In mammalian cells, mitochondrial fusion is primarily orchestrated by large dynamin-related GTPases termed mitofusin 1 (MFN1) and mitofusin 2 (MFN2), plus optic atrophy protein 1 (OPA1) [151,152]. MFN1 and MFN2 are transmembrane GTPases located in the outer mitochondrial membrane (OMM) and their primary function is to mediate the first step of mitochondrial fusion (fusion of the OMM), whereas OPA1 protein, a third GTPase of the dynamin family, is situated within the intermembrane space tightly associated with the inner mitochondrial membrane (IMM). Its primary function is to mediate fusion of the IMM. In addition, OPA1 has multiple roles, namely in maintaining cristae structure within the mitochondria, in maintaining inner membrane (IM) integrity and IM potential, and in preventing release of cytochrome c from the cristae [153]. The core components of mitochondrial fission (division) machinery are dynamin-related protein 1 (Drp1), mitochondrial fission 1 protein (Fis1), mitochondrial fission factor (Mff), and mitochondrial dynamin proteins of 49 and 51 kDa (MiD49/51) [154]. In addition to these mitochondrial components, the endoplasmic reticulum (ER) and actin cytoskeleton also contribute in mitochondrial division [154]. If the above fails, mitophagy is the next level of defense, ensuring the selective degradation of damaged mitochondria. The best-known pathway mediating mitophagy is the one that depends on the serine/threonine kinase PINK1 (phosphatase and tensin homolog induced putative kinase 1) and Parkin, an E3 ubiquitin ligase [155]. The former localizes to mitochondria while the latter resides in the cytosol. Under normal steady-state conditions, PINK1 undergoes a continuous import and sequential proteolysis cycle. This well-orchestrated process yields very low to undetectable levels of PINK1 on healthy mitochondria. PINK1 is stabilized specifically on the outer membrane of damaged mitochondria (e.g., due to depolarization or blocking mitochondrial import) flagging them for elimination. In particular, it activates Parkin’s E3 ubiquitin ligase activity, and recruits Parkin to the dysfunctional mitochondrion. Then, Parkin ubiquitinates outer mitochondrial membrane proteins and drives mitophagy to completion through a positive feedback-loop [156].

## 3. Cross-Talks between Impaired Mitostasis and Cellular Senescence

### 3.1. Impaired Mitochondrial Biogenesis and Cellular Senescence

Inefficient maintenance of the mitochondrial genome’s integrity due to defects/errors in the mtDNA replication machinery and/or failure in the repair of mtDNA damage leads to impaired mitochondrial biogenesis, mitochondrial dysfunction and bioenergetic failure of the cell. Despite the well-documented role of mutated mtDNA as a cause of different types of mitochondrial diseases [157], its impact as a driver of senescence is less investigated. Early studies, based on restriction enzyme analysis of mtDNA in fibroblasts undergoing replicative senescence, excluded the presence of deletions, insertions rearrangements, or single base changes [158]. Nevertheless, it was more recently shown in vitro that mtDNA-depleted cells display senescent phenotypes (resistance to cell death, increased SA-β-gal activity, lipofuscin accumulation), implicating the potential involvement of mtDNA damage in cellular senescence [159]. Indeed, current knowledge supports that all of the five nuclear-derived transcription factors that govern mitochondrial biogenesis, POLγ, PGC-1α, NRF-1/2, sirtuins, and TFAM have been somehow involved in cellular senescence [60].

Particularly, both the mitochondrial mass and the mRNA levels of PGC1α and NRF-1, were found to increase during replicative senescence in vitro [160]. This upregulation was attributed to *de novo* synthesis of the nuclear transcriptional factors as a compensatory response to increased ROS production and the impaired membrane potential [160]. On the other hand, overexpression of the transcriptional co-activator PGC-1α in human fibroblasts resulted in an increase of the mitochondrial encoded marker protein COX-II, consistent with the ability of PGC-1 to increase mitochondrial number, and accelerated the rate of cellular senescence [161].

In a model of OIS, oncogenic Ras induced multiple regulators of mitochondrial biogenesis, including NRF2a, PGC1α, PGC1β, and TFAM. Strikingly, even though the increased mRNA levels were documented two days after the induction of oncogenic *Ras*, the expression of these genes was even higher when the cells had established a full senescent state. Of note, newly formed mitochondria in *Ras*-senescent cells were dysfunctional, with compromised ATP generation and increased ROS, due to the continuous oncogenic stress [162]. At variance with these findings, in mice with dysfunctional telomeres, p53-dependent PGC1α and PGC-1β repression was shown to mediate cellular growth arrest [163,164]. PGC1 down-regulation resulted in reduced mitochondrial mass, impaired mitochondrial biogenesis, compromised OXPHOS and respiration with decreased ATP generation capacity, and decreased expression of ROS detoxifying enzymes. Enforced telomerase reverse transcriptase (TERT)—the catalytic subunit of the telomerase complex—or PGC-1α expression or germline deletion of p53 substantially rescues PGC network expression, mtDNA content and mitochondrial respiration.

In human cells and POLG^D257A^ mutated mice (i.e., a mutation in the proofreading domain of the mtDNA polymerase PolG), mitochondrial compromise due to genotoxic stress, caused by mtDNA depletion or accelerated rate of mtDNA mutations, has been associated with the induction of cellular senescence with a distinct secretory phenotype, one that lacks the IL-1-dependent inflammatory arm [38]. Importantly, elimination of the mitochondrial sirtuins SIRT3 and to a lesser extent SIRT5, but not other sirtuins, drove the senescent phenotype. In addition, while SIRT3 shRNA induced senescence in wild-type (WT) mouse embryonic fibroblasts (MEFs), MEFs from SIRT3 knockout mice did not senesce, thus suggesting that embryonic versus post-development acute loss of SIRT3 can have different effects [38]. Of great importance, mitochondrial dysfunction has been found to upset the balance of NAD^+^ (the oxidized form of nicotinamide adenine dinucleotide), a coenzyme that, besides its role in redox metabolism and cell signaling, also serves as a co-factor for sirtuins [165]. At the same time, both mitochondrial sirtuins and cytosolic NAD^+^ depletion have been implicated in the induction of premature senescence-like phenotype [38,166,167,168], therefore further underscoring the possible role of mitochondrial biogenesis impairment in cellular senescence through discoordination of energy metabolism [19].

Furthermore, in accordance with the notion that increased mitochondrial oxidative metabolism is a feature of cellular senescence, recent evidence suggests that the metabolic shift (i.e., increased mitochondrial oxidative metabolism) which characterizes cellular senescence, occurs in parallel with enhanced mitochondrial biogenesis [11,169]. Mechanistically, increased mitochondrial content was found to be regulated through a newly identified pathway, involving mechanistic target of rapamycin (mTOR)-dependent activation of PGC-1β, a key player in mitochondrial biogenesis [37]. It was also demonstrated that the reduction in mitochondrial content, by either mTORC1 inhibition or PGC-1β deletion, prevents senescence and attenuates SASP and ROS-dependent persistence of DDR [37].

Another cornerstone of mitochondrial biogenesis and maintenance of the mitochondrial genome’s integrity is the nuclear-encoded mitochondrial proteins. Notably, nuclear DNA is under the constant threat of oxidative damage due to ROS production, and from this point of view mitochondria seem to have a great impact as major contributors of oxidative stress. Nevertheless, the role of mitochondria extends far beyond the well-established impact of mitochondrial ROS as nuclear DNA damaging factors that activate a DDR and induce senescence [11,162]. Indeed, excessive mtDNA depletion can induce a reprogramming of nuclear gene expression patterns including genes involved in metabolism, stress response and growth signaling, termed “retrograde response” [170]. Dysfunctional mitochondria can actively secrete multiple forms of damage associated molecular patterns (DAMPS)—also known as mitochondrial alarmins—among of which are mtDNA and TFAM (the principal regulator of mtDNA transcription and stabilization). These molecules exit the mitochondrial compartment, enter the cytoplasm or the extracellular space, and bind to pattern recognition receptors (PRRs), such as toll-like receptors (TLRs) and NOD-like receptors (NLRs), thus activating the immune system and triggering a significant pro-inflammatory response [171,172]. Among others, cytosolic mtDNA can be recognized by and engage the cyclic GMP-AMP synthase (cGAS)-stimulator of interferon genes (STING) pathway which has been recently identified as a crucial regulator of senescence and the SASP [173]. Of great importance, cytochrome c, which under normal conditions is restricted within the mitochondrial intermembrane space where it functions as an electron carrier in the electron transport chain and as a scavenger of ROS, has also been identified as capable of serving as DAMP [171,172]. Indeed, cytochrome c seems to exert a biphasic role: apoptogenic or immunomodulatory. Upon stimuli, the release of cytochrome c into the cytoplasm is considered to be a critical event to facilitate the inflammation-free process of apoptosis, whereas when translocated extracellularly cytochrome c functions as a mitochondrial DAMP eliciting an inflammatory response [171,172]. Unfortunately, current knowledge regarding the spatiotemporal role of cytochrome c as a DAMP is still in its infancy and more studies are needed to elucidate the underlying molecular mechanisms.

It has also been demonstrated that a functional link between mitochondria and telomeres exists, suggesting a crosstalk between replicative senescence and mitochondria, with mitochondrial biogenesis holding a protagonist role [163]. Briefly, according to the proposed model, telomere-dysfunction-induced p53 represses the PGC network and compromises mitochondrial biogenesis. Specifically, in mice with dysfunctional telomeres, p53-mediated cellular growth arrest becomes activated, in turn repressing PGC-1α and PGC-1β, master regulators of metabolic and mitochondrial processes [163,164]. This results in reduced mitochondrial mass, impaired mitochondrial biogenesis, compromised OXPHOS and respiration with decreased ATP generation capacity, and down-regulated expression of ROS detoxifying enzymes. However, enforced telomerase reverse transcriptase (TERT)—the catalytic subunit of the telomerase complex—or PGC-1α expression or germline deletion of p53 substantially rescues PGC network expression, mtDNA content and mitochondrial respiration. Additionally, it has been proposed that telomerase protects mitochondria against oxidative stress through a telomere length-independent function. In particular, TERT is reversibly excluded from the nucleus upon both acute and chronic oxidative stress conditions, in a dose- and time-dependent manner, exported to the cytosol and colocalizes with/accumulates in mitochondria where it confers multilevel mitochondrial protection: decreases mitochondrial superoxide production and cell peroxide levels, enhances mitochondrial membrane potential, improves mitochondrial coupling, and reduces mtDNA damage, altogether suggesting improvement of the overall mitochondrial function [174]. In accordance, increased endogenous formation of ROS after continuous cultivation of endothelial cells was accompanied by both mitochondrial DNA damage and an export of nuclear TERT protein from the nucleus into the cytoplasm, followed by the onset of replicative senescence. Likewise, antioxidants delayed the onset of replicative senescence by counteracting the increased ROS production and preventing nuclear export of TERT protein [175]. Moreover, TERT overexpression suppressed retrograde response [170], which represents a characteristic feature of replicative senescence [29]. Of note, these finding are in discrepancy with earlier reports according to which ectopically expressed TERT in human fibroblasts under acute oxidative stress resulted in increased mtDNA damage [176,177,178].

Beyond ROS accumulation, mitochondrial dysfunction results in a decline in iron-sulfur cluster biogenesis which can stimulate nuclear genomic instability, which is manifested as a gradual slow of growth rate, a high frequency of cell death, or, surprisingly, cell-cycle arrest in the G1 phase and at a metabolically active status, reminiscing of senescence [179]. This cellular crisis would be expected to drive further decline in mitochondrial function via genotoxic activation of p53 and associated repression of PGC-1 family coactivators. Iron sulfur (Fe/S) clusters serve catalytic and structural functions in many cellular proteins, thus being involved in a wide variety of cellular processes such as enzymatic reactions, respiration, cofactor biosynthesis, ribosome biogenesis, regulation of gene expression, and DNA-RNA metabolism [180]. Noteworthy, in fibroblasts expressing oncogenic *Ras*, knocking down Rieske iron sulfur protein (RISP) of complex III leads to ROS production, a decrease in ATP synthesis, and activation of the AMPK pathway which triggers a robust senescent phenotype [162].

Another aspect of the involvement of mitochondrial genome instability in cellular senescence is its effect on the stem cell’s pool integrity. In mtDNA mutator mice, age-dependent accumulation of somatic mtDNA mutations has been suggested to affect stem cell homeostasis and eventually accelerates stem cell senescence. Potential mechanisms whereby mtDNA mutagenesis drives senescence in a stem cell population include loss of the mitochondrial membrane potential (MMP), blockage of metabolic shift during differentiation (from glycolysis to OXPHOS), imbalanced fusion and fission events (towards fission), abnormal mitophagy and/or autophagy, as well as ROS production [181].

### 3.2. Impaired Mitochondrial Dynamics and Cellular Senescence

The potential involvement of deregulated mitochondrial fusion, fission and mitophagy in cellular senescence has been suggested by a number of studies. Generally, in senescent cells, mitochondrial dynamics are considered to be strongly reduced [182]. Highly elongated mitochondria, accompanied with enhanced cristae structure and increased mitochondrial content, have been described during stress-induced premature senescence [23]. In line with this notion, the ultrastructural study of senescent cells of p21-inducible precancerous and cancerous cellular models (Li-Fraumeni and Saos-2 cell lines, respectively) studied previously by our group [183,184], revealed defective enlarged mitochondria in the majority of cells with perturbed morphology of cristae. Specifically, they were distributed mostly at the periphery of mitochondria or shaping circular formations, while in other mitochondria they were partially or totally lost (Figure 2). The above observations indicate dynamic remodeling of cristae responding to the metabolically needs of senescent cells or reflecting respiratory chain deficiency [185].

Moreover, some of the mitochondria were elongated (Figure 3) or branched (Figure 4) with abnormal distribution or partial loss of cristae indicating disturbance of mitochondrial dynamics.

As previously shown by Lee and colleagues [186], mitochondrial elongation has been associated with down-regulation of Fis1 along with an overall enhancement of fusion activity, as manifested by increased expression ratio(s) of Mfn proteins to fission modulators (Mfn > Drp1 and/or Mfn > Fis1). Direct induction of mitochondrial elongation by blocking the mitochondrial fission process was sufficient to develop a senescent phenotype with increased ROS production, whereas overexpression of Fis1 protein blocked the mitochondrial elongation and partially reversed the senescent phenotype. Remarkably, in case of simultaneous depletion of Fis1 and OPA1 (the critical component of mitochondrial fusion) or sequential depletion of OPA1 followed by Fis1 shRNA transfection, senescent-associated changes were significantly suppressed, and the cell proliferation rate was restored, even though mitochondria remained severely fragmented. This indicates that it is the fusion/fission imbalance that causes sustained mitochondrial elongation and not just the inhibition of mitochondrial fission *per se*, that triggers senescence-associated changes in Fis1 knockdown cells [186].

The formation of long and interconnected mitochondria in human endothelial cells (HUVECs) cultivated in vitro till they reached replicative senescence was associated with a reduced expression of Drp1 and Fis1 correlated with increased PINK1 mRNA levels [187]. The same mitochondrial architectural configuration is also adopted due to MARCH5 depletion that binds hFis1, Drp1 and Mfn2 [130,131,188]. The loss of MARCH5 facilitates mitochondrial elongation and interconnection either by suppression of Drp1-mediated mitochondrial or a marked increase in the steady-state levels of Mfn1, thus imposing a cellular stress which ultimately triggers cellular senescence [189]. Disruption of mitochondrial dynamics has been implicated in the induction of cellular senescence in human bronchial epithelial cells (HBEC). Mitochondrial fragmentation induced by knockdown of fusion proteins, OPA1 or MFN, was shown to boost mitochondrial ROS production and accelerate cellular senescence in HBEC exposed to cigarette smoke extract [190].

Taken together, in vitro studies show that senescent cells are typically associated with an overall shift toward more fusion events [31]. Whether mitochondrial elongation is causal to or epiphenomenon of cellular senescence has not yet been fully elucidated. Mitochondrial elongation could represent an energy-save attitude or even an adaptation to the impaired mitochondrial biogenesis that characterizes cellular senescence [145,187]. Others suggest that mitochondrial lengthening renders cells more resistant against apoptotic stimuli or autophagic degradation, thus facilitating cell viability [191,192,193,194,195,196]. Of interest, elongated and interconnected mitochondria of senescent endothelial cells exhibit a much higher threshold for stress-induced mitochondrial damage [187]. However, contradictory findings support that, in a longitudinal basis, prolonged elongated mitochondria ultimately result in higher production of intracellular ROS and diminished mitochondrial respiration activity [23].

Time-course analysis showed that mitochondrial population turnover is gradually declined in senescent cells in vitro and in vivo [197,198], as a consequence of reduced basal or induced autophagic activity, or due to lysosomal dysfunction and overload, which eventually overcome mitophagy capability [199]. This may partly explain the increased mitochondrial content of senescent cells [11,37].

It has been demonstrated that defective mitophagy and perinuclear build-up of damaged mitochondria is a critical contributor to the induction of cellular senescence in cigarette smoke extract-treated lung fibroblasts and small airway epithelial cells (SAECs). This is associated with impaired Parkin translocation and an exacerbation of mitochondrial ROS-induced DNA damage foci formation, due to cytoplasmic p53 accumulation [200]. Strikingly, in vitro experiments showed that Parkin overexpression was sufficient to induce mitophagy and repress accelerated cellular senescence in HBEC in response to cigarette smoke exposure, even in the setting of reduced PINK1 protein levels. Conversely PINK1 overexpression failed to recover impaired mitophagy caused by PRKN knockdown, suggesting that PRKN protein levels can be the rate-limiting factor in PINK1-PRKN-mediated mitophagy [201].

From the opposite point of view, cellular senescence directly contributes to dysregulated mitophagy that drives Senescence-Associated Mitochondrial Dysfunction (SAMD) [199]. Of great interest, SAMD is considered to be a major regulator of the senescent phenotype, especially of the SASP, thus contributing to the development and stability of the senescent cell cycle arrest [11,38,202].

Furthermore, the regulation and functional role of mitophagy in cellular senescence appears also to be related to changes in general autophagy, even though things are less clear. By removing damaged macromolecules or organelles, autophagy prevents garbage catastrophe, thus exerting an anti-senescence role. However, on a short-term basis, autophagy facilitates the synthesis of senescence-associated secretory proteins, thus suggesting to be a pro-senescence mechanism [203]. It was demonstrated that autophagy impairment with lysosomal and mitochondrial dysfunction is crucial for oxidative stress-induced cell senescence [27]. On the contrary, targeted mitochondrial damage due to oxidative stress-upregulated autophagy factors LC3B, ATG5 and ATG12, enhanced mitophagy and prevented senescence [204].

## 4. Future Perspectives

Intriguingly, the onset of the senescent phenotype is not always beneficial. Short-term accumulation of senescent cells has a positive outcome in embryonic development, tissue repair, and cancer prevention. On the other hand, its chronic persistence (chronic senescence) leads to detrimental results, such as aging and age-related pathologies [205]. Respectively, impaired mitochondrial function as well as cellular senescence are both implicated in aging and age-related pathologies such as cancer, neurodegenerative and cardiovascular diseases [206,207]. Except for the mitochondrial free radical theory of aging which highlights the accumulation of mitochondrial oxidative damage (due to progressive mitochondrial dysfunction and increased production of ROS) as the driving force of age-related phenotypes, the current view supports the notion that aging is, among other causes, the result of generalized impaired mitochondrial bioenergetics that cause global cellular damage [119,208]. In addition, cellular senescence has also been recognized as a hallmark of aging; although in young organisms, cellular senescence acts as a failsafe program to prevent the propagation of damaged cells, the deficient clearance of senescent cells in aged tissues results in accumulation of senescent cells which exert deleterious effects and jeopardize tissue homeostasis [208].

This also has therapeutic perspectives. Elimination of senescent cells in a selective manner over normal cells has been proven to prevent or delay tissue dysfunction and to maximize healthy lifespan as exemplified in progeroid animal models [97]. Moreover, a new research field has opened up, where strategies can be designed to reduce the burden of senescent cells in an organism and thus contribute to the treatment of pathological conditions and age-related abnormal conditions. Given that mitochondrial dysfunction—at least partly—drives senescence, targeting mitochondrial dysfunction emerges as a potential therapeutic strategy to counteract the negative impact of chronic senescence. In this regard, resveratrol, a polyphenol which has been shown to exert immunomodulatory, anti-inflammatory and antioxidative effects, with an ability to prolong lifespan and protect against age-related disorders in different animal models, has gained attention as a potential senolytic agent [209]. It has been demonstrated that resveratrol improves mitochondrial function and protects against metabolic disease by inducing PGC-1a and SIRT1 activity [210]. Moreover, it was recently reported the role for mitochondria in specific elimination of senescent cells using mitochondria-targeted tamoxifen (MitoTam), based on the capacity of non-proliferating non-cancerous cells to withstand oxidative insult induced by OXPHOS inhibition [211].

SASP action is considered to be the major modulating factor of the bimodal behavior that senescent cells exert. Therefore, mitochondrial-targeted interventions for selective inhibition of the SASP components can elicit anti-senescent effects. As previously mentioned, senescent cells exhibit impaired mitochondrial biogenesis and metabolic shifts, namely a decrease in NAD^+^ and an increase in AMP and ADP. These changes have been shown to contribute to both the senescent cell cycle arrest as well as the regulation of the SASP via multiple signaling pathways. The core idea is that mitochondrial ablation upon induction of senescence, selectively inhibits common pro-inflammatory and pro-oxidant aspects of the senescent phenotype, while preserving the cell cycle arrest, which in specific context (e.g., late stage of tumorigenesis) is desirable. In this regard, possible mechanisms whereby mitochondria that have abolished normal function are implicated in SASP regulation include: (a) mTOR activation due to sustained DDR which promotes PGC1-β dependent biogenesis of new, yet dysfunctional, mitochondria that further increase ROS production, thus replenishing DDR through a positive feedback-loop, (b) AMPK activation (due to increased AMP/ATP and ADP/ATP ratios) which in turn activates p53 and subsequently stabilizes p16 and p21, thus promoting cell cycle arrest, (c) low NAD^+^-driven inhibition of poly-ADP ribose polymerases (PARPs) which are dispensable for DNA repair after genotoxic stress, (d) low NAD+-driven inactivation of sirtuins, which normally serve as inhibitors of NF-kB activity and transcriptional repressor of SASP genes, (e) initiation of an innate immune response due to cytosolic exit of damaged mtDNA molecules that exert pro-inflammatory effects [18]. Moreover, recent studies indicate that mTOR inhibition contributes to reduction of the SASP by decreasing translation of the proteins interleukin-1 α (IL-1A) and MAP kinase-activated protein kinase 2 (MAPKAPK2) or via reduction of mitochondrial biogenesis and ROS-dependent persistence of a DDR [37,212,213].

In addition, activation of autophagy by inhibition of mTORC was shown to efficiently suppress senescence phenotypes in a number of studies [27,37,202]. Of great importance, the introduction of senolytic strategies is a relative novel and unexplored field. A high level of caution is needed since new findings are coming into light underscoring possible undesirable side effects. For example, a category of senolytic drugs that function as inhibitors of the anti-apoptotic BCL-2 family proteins has been shown to induce a minor mitochondrial outer membrane permeabilization (miMOMP) due to limited caspase activation, not sufficient to induce apoptosis, yet capable of causing increased DNA damage and genomic instability, even in neighboring non-senescent cells [18,214]. Of clinical relevance, a recently developed chemically modified mitochondria-targeted doxorubicin derivative was shown to be less cardiotoxic and more effective than doxorubicin, against drug-resistant tumor cells overexpressing P-glycoprotein [215]. Even though the role of mitochondria in the various modes of cell death and cell physiology has been well known, their involvement in cellular senescence has only recently started to be elucidated. At the moment, a thorough understanding of the mechanisms governing the bidirectional connection between perturbations in mitochondrial homeostasis and cellular senescence is missing. Novel methodologies for the detection of cellular senescence and new technologies applied to the analysis of mitochondrial biochemistry continue to be developed, thus facilitating our understanding of these multifaceted organelles and elucidating the interplay between mitochondria and cellular senescence [216,217].

## Figures and Tables

**Figure 1 cells-08-00686-f001:**
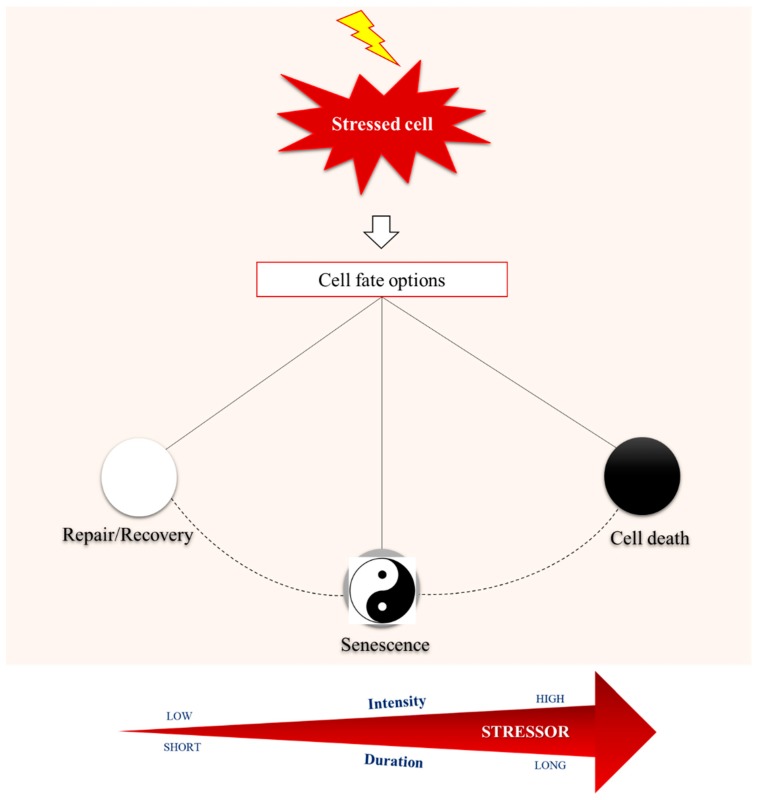
Maintaining homeostasis is the cornerstone for cells’ normal function, ensuring organismal physiology. Intriguingly, cells are constantly exposed to intrinsic and extrinsic stressors that jeopardize cellular integrity and activate a variety of response modules, through complex and highly sophisticated biochemical networks. Depending on the intensity and duration of the stressor, cellular response mechanisms either manage to neutralize the adverse effects of stress, thus achieving complete recovery and survival, or lead to death in case of non-repairable damage. Between these two opposite outcomes reminiscent of the swinging of a pendulum, cellular senescence enters the scene.

**Figure 2 cells-08-00686-f002:**
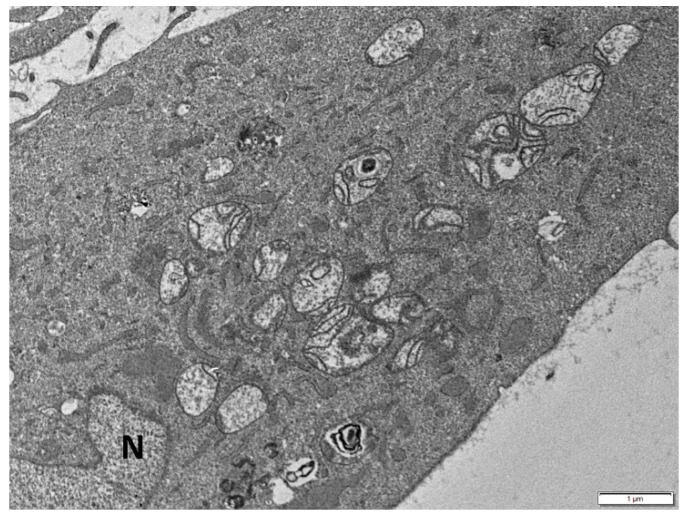
Senescent cell with enlarged mitochondria with disturbed morphology of cristae distributed mostly at their periphery, forming circular constructions, or partially lost. N: nucleus. Scale bar: 1 μm.

**Figure 3 cells-08-00686-f003:**
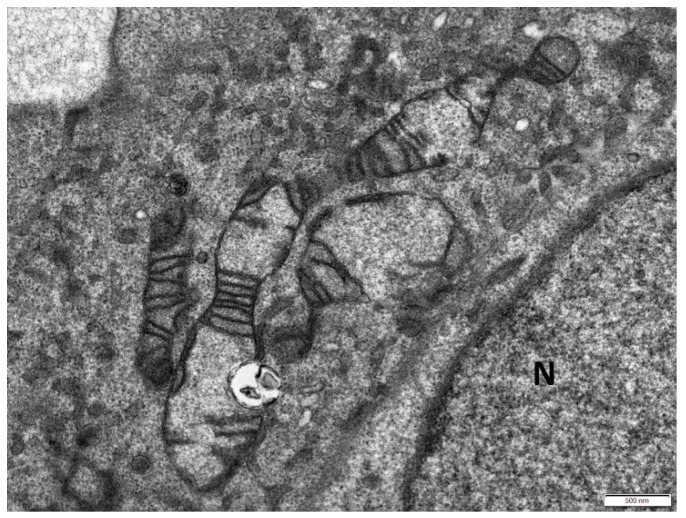
Elongated mitochondria in the cytoplasm of a senescent cell with partial loss of cristae. N: nucleus. Scale bar: 500 nm.

**Figure 4 cells-08-00686-f004:**
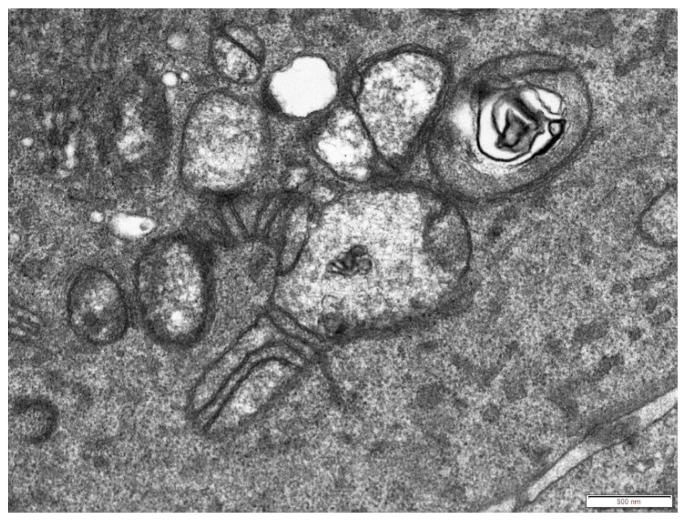
Branched mitochondrion in the cytoplasm of a senescent cell with partial loss of cristae. Scale bar: 500 nm.

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
