# Peer review of "Mitochondrial Homeostasis and Cellular Senescence"

_cells, 2019, doi:10.3390/cells8070686_

Round 1
Reviewer 1 Report
This review provides an overview of homeostatic mechanisms operating within mitochondria and a comprehensive insight into the interactions between cellular senescence and mitochondrial dysfunction.
It is a very interesting this review, however there are some concerns regarding I have some comments as follows:
1. Does mitochondrial ROS aggravate cellular senescence of senescent cells? You should describe that mitochondrial ROS can aggravate cellular senescence.
2. Dysfunctional mitochondria secrete damage associated molecular patterns (DAMPs), such as mtDNA etc. These activate the inflammasome and enhance the secretion of inflammatory cytokines. On the other hand, when DAMPs is secreted, is apoptosis-inducing factor cytochrome c secreted together from Dysfunctional mitochondria? If cytochrome c is not secreted simultaneously with DAMPs, please you describe this.
Author Response
Comments and Suggestions for Authors
“This review provides an overview of homeostatic mechanisms operating within mitochondria and a comprehensive insight into the interactions between cellular senescence and mitochondrial dysfunction.
It is a very interesting this review, however there are some concerns regarding I have some comments as follows:
1. Does mitochondrial ROS aggravate cellular senescence of senescent cells? You should describe that mitochondrial ROS can aggravate cellular senescence.
2. Dysfunctional mitochondria secrete damage associated molecular patterns (DAMPs), such as mtDNA etc. These activate the inflammasome and enhance the secretion of inflammatory cytokines. On the other hand, when DAMPs is secreted, is apoptosis-inducing factor cytochrome c secreted together from Dysfunctional mitochondria? If cytochrome c is not secreted simultaneously with DAMPs, please you describe this.”
Response to Reviewer #1
1. Comment/Suggestion
“Does mitochondrial ROS aggravate cellular senescence of senescent cells? You should describe that mitochondrial ROS can aggravate cellular senescence.”
Response: Based on your accurate comment, we added the following sentence (highlighted in red) (Lines 83-85): “Senescent cells are characterized by increased production of reactive oxygen species (ROS), mainly attributed to dysfunctional mitochondria [26]. Indeed, in already senescent cells, mitochondrial ROS can aggravate cellular senescence by enhancing the DNA damage and the DNA damage response signaling pathway (DDR) [11]. Noteworthy, mitochondrial deoxyribonucleic acid (mtDNA) is highly vulnerable to ROS due to proximity to the generation site, whilst damaged mtDNA in turn, impairs OXPHOS function, thus further enhancing ROS release [17].”
Let us kindly mention that the role of mitochondrial ROS as “potentiating” or triggering factors of cellular senescence is mentioned several times throughout the manuscript, however our scope when writing this review was to shift the focus regarding mitochondria and senescence beyond the role of ROS and oxidative stress.
2. Comment/Suggestion
“Dysfunctional mitochondria secrete damage associated molecular patterns (DAMPs), such as mtDNA etc. These activate the inflammasome and enhance the secretion of inflammatory cytokines. On the other hand, when DAMPs is secreted, is apoptosis-inducing factor cytochrome c secreted together from Dysfunctional mitochondria? If cytochrome c is not secreted simultaneously with DAMPs, please you describe this.”
Response: Taking into consideration the valuable comment, we added the following text (Lines 434-443): “Of great importance, cytochrome c, which under normal conditions is restricted within the mitochondrial intermembrane space where it functions as an electron carrier in the electron transport chain and as a scavenger of ROS, has been also identified as capable of serving as DAMP [171,172]. Indeed, cytochrome c seems to exert a biphasic role: apoptogenic or immunomodulatory. Upon stimuli, the release of cytochrome c into the cytoplasm is considered to be a critical event to facilitate the inflammation-free process of apoptosis, whereas when translocated extracellularly cytochrome c functions as a mitochondrial DAMP eliciting an inflammatory response [171,172]. Unfortunately, current knowledge regarding the spatiotemporal role of cytochrome c as a DAMP is still in its infancy and more studies are needed to elucidate the underlying molecular mechanisms.”
Reviewer 2 Report
Date 25-06-2019
Revision of Manuscript: “Mitochondiral homeostasis and cellular senescence” by Vasileiou et al. for the Journal “Cells” of MDPI publisher.
The article is a well written in deep review of the role of mitochondria metabolism and dynamic and their possible role in cellular senescence. The article cite over 200 references showing that different cellular senescence programs affect mitochondria, but also that mitochondria alterations can modify the senescent program itself.
Cellular senescence metabolism is an expanding field of investigation and the review will be a useful and timely tool for researcher who aim to focus in this research field.
I have only noticed few minor oversights:
Page 4, line 137: d-loop, should be written always D-loop for consistency.
Page 5, line 166: One word is missing in the sentence: “At least in some tissues, Polβ interacts WITH nucleoid….”
Page 10, line 399: NAD+ is a coenzyme, not an enzyme.
It is my usual habit to disclose my identity to the authors: Antonello Lorenzini.
Author Response
Comments and Suggestions for Authors
Revision of Manuscript: “Mitochondrial homeostasis and cellular senescence” by Vasileiou et al. for the Journal “Cells” of MDPI publisher.
The article is a well written in deep review of the role of mitochondria metabolism and dynamic and their possible role in cellular senescence. The article cite over 200 references showing that different cellular senescence programs affect mitochondria, but also that mitochondria alterations can modify the senescent program itself.
Cellular senescence metabolism is an expanding field of investigation and the review will be a useful and timely tool for researcher who aim to focus in this research field.
I have only noticed few minor oversights:
Page 4, line 137: d-loop, should be written always D-loop for consistency.
Page 5, line 166: One word is missing in the sentence: “At least in some tissues, Polβ interacts WITH nucleoid….”
Page 10, line 399: NAD+ is a coenzyme, not an enzyme.
Response to Reviewer #2
Comment/Suggestion:
Page 4, line 137: d-loop, should be written always D-loop for consistency.
Response: Taking into consideration your valuable notification, we proceeded with the relevant corrections wherever needed: “…In addition, D-loop has been implicated in protein recruitment…” (Line 144)
Comment/Suggestion:
Page 5, line 166: One word is missing in the sentence: “At least in some tissues, Polβ interacts WITH nucleoid….”
Response: We completed the missing word as follows: “…At least in some tissues, Polβ interacts with nucleoid proteins such as TWINKLE helicase…” (Line 174)
Comment/Suggestion:
Page 10, line 399: NAD+ is a coenzyme, not an enzyme.
Response: Taking into consideration your valuable notification, we made the relevant correction as follows: “…mitochondrial dysfunction has been found to upset the balance of NAD+ (the oxidized form of nicotinamide adenine dinucleotide), a coenzyme that besides its role in redox metabolism…” (Line 404)
Reviewer 3 Report
This is a nice review paper on crosslinks between cellular senescence and mitochondrial function. The authors overviewed the major controlling factors for senescence and mitochondria yet I'd like to see a paragraph or two about possible interference of this association in specific pathologies (including aging) as well as possible senescence-interefering mitochondria-specific agents (e.g. resveratrol, doxorubicin).
Author Response
Comments and Suggestions for Authors
English language and style
( ) Extensive editing of English language and style required
( ) Moderate English changes required
(x) English language and style are fine/minor spell check required
( ) I don't feel qualified to judge about the English language and style
“This is a nice review paper on crosslinks between cellular senescence and mitochondrial function. The authors overviewed the major controlling factors for senescence and mitochondria yet I'd like to see a paragraph or two about possible interference of this association in specific pathologies (including aging) as well as possible senescence-interefering mitochondria-specific agents (e.g. resveratrol, doxorubicin).”
Response to Reviewer #3
Comment/Suggestion:
English language and style; English language and style are fine/minor spell check required
Response: We made considerable effort to improve English language and to eliminate any grammatical and typos errors.
Comment/Suggestion:
“…The authors overviewed the major controlling factors for senescence and mitochondria yet I'd like to see a paragraph or two about possible interference of this association in specific pathologies (including aging) as well as possible senescence-interefering mitochondria-specific agents (e.g. resveratrol, doxorubicin).”
Response: We are very much thankful to the reviewer for the comment. Based on this suggestion, we added the following data in the Future Perspectives section (Lines 592-601, 606-617, 649-651):
“Intriguingly, the onset of the senescent phenotype is not always beneficial. Short term accumulation of senescent cells has a positive outcome in embryonic development, tissue repair, and cancer prevention. On the other hand, its chronic persistence (chronic senescence) leads to detrimental results, such as aging and age-related pathologies [205]. Respectively, impaired mitochondrial function as well as cellular senescence are both implicated in aging and age-related pathologies such as cancer, neurodegenerative and cardiovascular diseases [206,207]. Except for the mitochondrial free radical theory of aging which highlights the accumulation of mitochondrial oxidative damage (due to progressive mitochondrial dysfunction and increased production of ROS) as the driving force of age-related phenotypes, current view supports the notion that aging is, among other causes, the result of generalized impaired mitochondrial bioenergetics that cause global cellular damage [208]. In addition, cellular senescence has also been recognized as a hallmark of aging; although in young organisms, cellular senescence acts as a failsafe program to prevent the propagation of damaged cells, the deficient clearance of senescent cells in aged tissues results in accumulation of senescent cells which exert deleterious effects and jeopardize tissue homeostasis [208].
This has also therapeutic perspectives. Elimination of senescent cells in a selective manner over normal cells has been proven to prevent or delay tissue dysfunction and to maximize healthy lifespan as exemplified in progeroid animal models [97]. Moreover, a new research field has opened up, where strategies can be designed to reduce the burden of senescent cells in an organism and thus contribute to the treatment of pathological conditions and age-related abnormal conditions. Given that mitochondrial dysfunction-at least partly-drives senescence, targeting mitochondrial dysfunction emerges as a potential therapeutic strategy to counteract the negative impact of chronic senescence. In this regard, resveratrol, a polyphenol which has been shown to exert immunomodulatory, anti‐inflammatory and antioxidative effects, with an ability to prolong lifespan and protect against age‐related disorders in different animal models has gained attention as a potential senolytic agent [209]. It has been demonstrated that resveratrol improves mitochondrial function and protects against metabolic disease by inducing PGC-1a and SIRT1 activity [210]. Moreover, it was recently reported the role for mitochondria in specific elimination of senescent cells using mitochondria-targeted tamoxifen (MitoTam), based on the capacity of non-proliferating non-cancerous cells to withstand oxidative insult induced by OXPHOS inhibition [211].
… Of clinical relevance, a recently developed chemically modified mitochondria-targeted doxorubicin derivative was showed to be less cardiotoxic and more effective than doxorubicin, against drug-resistant tumor cells overexpressing P-glycoprotein [215]…”.